# Preparation, In Vivo and In Vitro Release of Polyethylene Glycol Monomethyl Ether-Polymandelic Acid Microspheres Loaded *Panax Notoginseng Saponins*

**DOI:** 10.3390/molecules24102024

**Published:** 2019-05-27

**Authors:** Yi He, Hongli Li, Xiangyu Zheng, Mingwei Yuan, Renyu Yang, Minglong Yuan, Cui Yang

**Affiliations:** National and Local Joint Engineering Research Center for Green Preparation Technology of Biobased Materials, Yunnan Minzu University, Kunming 650500, China; heyi_sichuan@163.com (Y.H.); honglili@vip.163.com (H.L.); zhengxiangyu1993@163.com (X.Z.); yuanmingwei@163.com (M.Y.); yangrenyu1995@163.com (R.Y.)

**Keywords:** *Panax notoginseng saponins*, microspheres, sustained-release, biocompatibility, anti-inflammatory, anti-cancer

## Abstract

In order to enrich the types of *Panax notoginseng* saponins (PNS) sustained-release preparations and provide a new research idea for the research and development of traditional Chinese medicine sustained-release formulations, a series of *Panax notoginseng saponins* microspheres was prepared by a double emulsion method using a series of degradable amphiphilic macromolecule materials polyethylene glycol monomethyl ether-polymandelic acid (mPEG-PMA) as carrier. The structure and molecular weight of the series of mPEG-PMA were determined by nuclear magnetic resonance spectroscopy (^1^ HNMR) and gel chromatography (GPC). The results of the appearance, particle size, drug loading and encapsulation efficiency of the drug-loaded microspheres show that the mPEG10000-PMA (1:9) material is more suitable as a carrier for loading the total saponins of *Panax notoginseng*. The particle size was 2.51 ± 0.21 μm, the drug loading and encapsulation efficiency were 8.54 ± 0.16% and 47.25 ± 1.64%, respectively. The drug-loaded microspheres were used for in vitro release and degradation experiments to investigate the degradation and sustained release behaviour of the drug-loaded microspheres. The biocompatibility of the microspheres was studied by haemolytic, anticoagulant and cytotoxicity experiments. The pharmacological activity of the microspheres was studied by anti-inflammatory and anti-tumour experiments. The results showed that the drug-loaded microspheres could be released stably for about 12 days and degraded within 60 days. At the same time, the microspheres had good biocompatibility, anti-inflammatory and anti-tumour activities.

## 1. Introduction

*Panax notoginseng* saponins (PNS) are the main components extracted from the traditional Chinese medicine *Panax notoginseng* [1]; they have wide applications for the treatment of cardiovascular and cerebrovascular diseases and possess anti-inflammatory and anti-tumour activities, among others [2,3,4,5,6,7]. However, long-term administration is required for patients with many cardiovascular and cerebrovascular diseases, chronic inflammation and malignant tumours. PNS have limitations in humans, such as a short half-life, low bioavailability and easily damaged active monomers due to the acids/enzymes in the stomach environment [8] and thus patients are required to frequently take large doses of the medication to maintain the effective blood drug levels. At the same time, it is easy to cause accumulation of drug concentration to cause other adverse reactions. The development of an efficient drug loading and sustained release system is the key to solving these problems. However, very few reports on the sustained release of PNS have been published [9].

The use of degradable polymer materials as drug carriers to deliver drugs is one of the directions in the development of drug delivery systems [10,11,12,13,14]. The use of amphiphilic biodegradable polymers as carriers to load drugs and prepare drug-loaded micelles or microspheres increases the stability and hydrophilicity of the drug, increases the circulation time of the drug in the blood, improves the bioavailability of the drug and at the same time, also reduces toxic side effects [15,16,17,18]. Currently, the sustained-release microspheres containing PNS mainly use chitosan [19], albumin [20] and *β*-cyclodextrin [21] as drug carriers. However, drug-loaded microspheres composed of chitosan and *β*-cyclodextrin have issues with hydrophilicity and unsatisfactory drug loading and drug release. Albumin microspheres also have problems with unsatisfactory sustained release effect, high cost and limited applications.

As a lipophilic polymer, polymandelic acid (PMA) exhibits good biodegradability and biocompatibility, similar to polylactic acid (PLA), and has great potential value for applications as a drug carrier. Polyethylene glycol monomethyl ether (mPEG) displays good hydrophilicity and is widely used to modify the lipophilic drug carrier PLA to improve its hydrophilicity [22,23,24,25,26,27,28,29,30,31]. Copolymerization of mPEG with mandelic acid O-carboxyanhydride (OCA) improves the hydrophilicity of PMA and produces the amphiphilic carrier material mPEG-PMA. Therefore, in this paper, mPEG-PMA was used as the carrier and a water-in-oil-in-water (W/O/W)-type double emulsion was prepared using the solvent emulsification volatilization method [32]. The organic solvent was further removed through solid-liquid separation, washing and drying to obtain solid microspheres. At the same time, the morphology and particle size of the drug-loaded microspheres were characterized using scanning electron microscopy (SEM) and a laser particle size analyser. The drug loading and encapsulation efficiencies were determined based on a standard curve established using the absorbance values obtained through spectrophotometry. The results of the measurements were comprehensively evaluated to determine the optimal drug-loaded microspheres. Finally, the degradation and sustained release behaviours of the drug-loaded microspheres were investigated using in vitro release and degradation experiments. The biocompatibility of the microspheres was evaluated using haemolytic, anticoagulant and cytotoxicity experiments. The pharmacological activity of the microspheres was investigated by performing anti-inflammatory and anti-tumour experiments.

## 2. Results and Discussion

### 2.1. Characterization Results and Analysis of Copolymer mPEG-PMA

Figure 1 shows the ^1^H nuclear magnetic resonance (NMR) spectrum of the mPEG-PMA copolymer. Peak a with the chemical shift δ = 3.43 ppm is attributed to the characteristic peak of the H proton on the CH_3_O- group in mPEG at the end of the copolymer. Peak b at δ = 3.61–3.72 ppm is the methylene group in the mPEG block in the copolymer. Peak c with the chemical shift δ = 5.83−6.21 ppm is attributed to the characteristic peak of the methylene H proton in the PMA block of the copolymer. Peaks (d+e) with the chemical shift δ = 6.91−7.54 ppm are the characteristic peaks of the methylene H proton of the benzene ring in PMA.

Table 1 shows the results of the gel chromatography (GPC) assessment of different block copolymers obtained by the copolymerization of mPEG with different molecular weights and Mac-OCA, and the copolymerization of the same molecular weights of mPEG and Mac-OCA at different feed ratios (mass ratio). The molecular weight of the copolymer increased as the molecular weight of the reactant material mPEG increased when the ratio of mPEG to Mac-OCA was the same. When the same molecular weight of mPEG was used, the molecular weight of the copolymer decreased as the feed ratio of mPEG/Mac-OCA increased because the number of hydroxyl groups in the copolymer increases as the ratio of mPEG in the copolymer increases, decreasing the chain length of the copolymer and the molecular weight. The polydispersity indexes (PDIs) of the copolymers were all less than 1.21, indicating a narrow molecular weight distribution of the polymer and a uniform molecular weight of the polymer.

The morphologies of the aforementioned PNS microspheres, loaded with different materials obtained using SEM, are shown in Figure 2, in which the drug-loaded microspheres from samples a to e correspond to (**a**) to (**e**) in the figures, respectively. As shown in the figure, the microspheres displayed good sphericity with a smooth surface and little sticky bonding phenomenon, indicating that the microspheres have good sphere formation and encapsulation properties, which are favourable for sustained and stable release. 

Figure 3 shows the particle size distribution of the drug-loaded microspheres composed of different materials. As shown in the figure, mPEG2000-PMA (1:9) microspheres had an average particle size distribution of 0.68 ± 0.12 μm, mPEG5000-PMA (1:9) microspheres had an average particle size distribution of 1.52 ± 0.18 μm, mPEG10000-PMA (1:9) microspheres had an average particle size distribution of 2.51 ± 0.21 μm, mPEG10000-PMA (1:7) microspheres had an average particle size distribution of 1.91 ± 0.11 μm and mPEG10000-PMA (1:5) microspheres had an average particle size distribution of 1.78 ± 0.16 μm. Based on the morphology and the particle size distribution, the particle size of the microspheres increased as the molecular weight of the mPEG increased, and decreased as the ratio of the mPEG block increased. The particle size of the drug-loaded microspheres is less than 3 μm and can be administered by various administration methods such as oral administration, nasal mucosal administration, intra-articular injection and so forth. The administration method is flexible.

### 2.2. Determination and Analysis of the Encapsulation Rate and Percentage of Drug Loading in the PNS Microspheres

The optical density (OD) values of the PNS standard solutions with known concentrations served as the ordinate and the known PNS concentration C (μg/mL) was used as the abscissa to establish the following standard curve: Y = 0.3019X + 0.0642, *R^2^* = 0.9992. PNS displayed a good linear relationship in the range of 2.5–250 μg/mL. The measured OD value of the PNS released from the PNS-loaded microspheres was substituted into the standard curve and the content of PNS contained in the microspheres was calculated to enable the subsequent calculation of the drug loading and encapsulate rates of the PNS-loaded microspheres. A detailed description of the results is presented in Table 2. The drug loading and encapsulation efficiency of the microspheres increased with the increase of the molecular weight of the carrier material and the drug-loaded microspheres with the largest molecular weight of mPEG10k-PMA (1:9) had better encapsulation efficiency and drug loading. The reason is that as the molecular weight increases, the chain length of the copolymer increases and the larger the "core" diameter in the typical "core-shell" structure formed by the hydrophilic end and the hydrophobic end wrapping the PNS, the higher encapsulation efficiency and drug loading.

### 2.3. In Vitro Analysis of Drug Release from the Microspheres

Figure 4 shows the in vitro release profile of PNS-loaded microspheres. The sustained release was stable, and an obvious burst release phenomenon was not observed. Additionally, the rate of drug release was positively correlated with the proportion of hydrophilic blocks in the material. 36.4% of the drug inside the mPEG10000-PMA microspheres (1:9) was dissolved in the first 12 days of release. The cumulative release of the drug reached 65.9% at day 48. No burst release phenomenon was observed during the release process, which is a very advantageous feature for drug delivery systems and is beneficial for the sustained release of the drug-loaded microspheres in the human body within a certain time range, prolonging the retention time and effective blood drug concentration of PNS in the human body. By comprehensively considering the appearance, particle size, drug loading rate, encapsulation rate and in vitro release effect, mPEG10000-PMA (1:9) drug-loaded microspheres were selected for assessments of biocompatibility and pharmacological activity.

### 2.4. In Vitro Analysis of Microsphere Degradation

Figure 5 shows the pH, Mn and weight loss change curves produced during the degradation of the microspheres. The trends of the three curves were basically the same. Within 60 days of continuous degradation of the microspheres, the acid anhydride bonds of the polyanhydride were broken and the molecular weight and mass of the microspheres also decreased. The change in the pH of the degradation medium was significant, from 7.40 to 3.41. However, the decrease in the pH of the degradation medium was mainly due to the presence of easily hydrolysed anhydride bonds in the mPEG-PMA microspheres. When a large amount of degradation medium contacted the microsphere matrix, the anhydride bonds in the microsphere carrier material broke into fragments or shorter oligomers, resulting in the production of the small molecule mandelic acid, which decreased the pH of the degradation medium. After 60 days, the changes in pH, Mn and weight loss tended to plateau, indicating that the degradation was basically complete.

### 2.5. Analysis of Haemolysis Induced by the Microspheres

Figure 6 shows the results of the assessment of the haemolytic activity of PNS, mPEG-PMA and PNS-loaded microspheres in the concentration range of 0.4–40 μg/mL. PNS, mPEG-PMA and PNS-loaded microspheres all exhibited low haemolytic activity within the set concentration range and the haemolysis rate increased with increasing concentrations but was < 5%, which meets the regulation of the International Standardization Organization (ISO), indicating that the drug-loaded microspheres are safe for contact with blood at a concentration ≤ 40 μg/mL and the drug-loaded microspheres have good biocompatibility [33,34].

### 2.6. Analysis of Anticoagulant Activity of the Microspheres

Figure 7 shows the results of anticoagulant index test of PNS, mPEG-PMA and the drug-loaded microspheres. PNS, mPEG-PMA and drug-loaded microspheres all exhibited certain anticoagulant properties and the anticoagulant indexes of the three preparations increased as the concentration increased within a set concentration range (0.4–40 μg/mL), indicating that the drug-loaded microspheres do not affect the anticoagulant properties of PNS itself and can be used as a PNS antithrombotic agent.

### 2.7. Analysis of the Cytotoxicity of the Microspheres

Figure 8 shows the results of the assessment of the cytotoxicity of PNS-loaded mPEG-PMA microspheres, mPEG-PMA and PNS toward HL7702 human normal liver cells. Even at the highest tested concentration of 40 μg/mL, PNS, mPEG-PMA and the drug-loaded mPEG-PMA microspheres were not cytotoxic to HL-7702 human normal liver cells because the relative growth rates of cells were greater than 100%. Thus, at a concentration of ≤ 40 μg/mL, PNS, the microspheres and the drug-loaded microspheres are not toxic to HL-7702 human normal liver cells.

### 2.8. Analysis of the Anti-Inflammatory Activity of the Microspheres

Table 3 shows the results of the assessment of the anti-inflammatory activity of PNS microspheres using the *p*-xylene-induced ear swelling model in mice. High, medium and low doses of PNS-loaded microspheres exhibited good anti-inflammatory activity. Although the anti-inflammatory activity of the drug-loaded microspheres was not as good as PNS, ASP and *Panax notoginseng* slices over a short time period, the drug-loaded microspheres stably released the drug for a long time, had a longer half-life and maintained the minimum effective concentration for a longer period. Therefore, in terms of long-term efficacy and bioavailability, the drug-loaded microspheres function better.

Table 4 shows the results of the assessment of the anti-inflammatory activity of the PNS-loaded microspheres using the egg white-induced paw swelling model in rats. High, medium and low doses of the drug-loaded microspheres all showed good anti-inflammatory activity. The high dose exerted the best anti-inflammatory effect, followed by the middle dose and the low dose, indicating that the anti-inflammatory activity of the drug-loaded microspheres was positively correlated with the dose.

### 2.9. Analysis of the Anti-Tumour Activity of the Microspheres

Figure 9 shows the results of the in vitro assay of the anti-tumour activity of PNS. The anti-tumour activity of the drug-loaded microspheres increased as the concentration increased in the range of 0.4–40 μg/mL. The anti-tumour activity of the medium and low doses was not much different from the original drug. The anti-tumour activity of the high-dose drug-loaded microspheres was weaker than PNS. However, the PNS-loaded microspheres continuously and stably released the drug with a long half-life; therefore, the overall performance was better than the original PNS drug.

## 3. Materials and Methods 

### 3.1. Materials

4-Dimethylaminopyridine (DMAP) was purchased from Aladdin Reagent Co., Ltd. (Shanghai, China). Tetrahydrofuran (THF) was purchased from Tianjin Damao Chemical Reagent Factory (Tianjin, China). *Panax notoginseng saponins* (provided by Yunnan University of Traditional Chinese Medicine) and polyvinyl alcohol (PVA, Mw = 75,000 Da, 88% alcoholysis) were purchased from Shanghai Jingchun Reagent Co., Ltd.; mPEG with various molecular weights was purchased from Shanghai Seebio Biotechnology Inc. (Shanghai, China). Kunming mice, Sprague-Dawley (SD) rats and male rabbits were provided by Kunming Medical University, China. All animal experiments were conducted according to the principles of the National Institute of Health (NIH) Guide for the Care and Use of Laboratory Animals and were approved by the ethics committee for laboratory animal care and use of the Institute of Materia Medica, and CAMS & PUMC.HL-7702 cells (normal human liver cells) were purchased from Procell Life Science & Technology Co., Ltd. (Wuhan, China). High-glucose 1640 medium was provided by HyClone. The 3-(4,5-dimethyl-2-thiazolyl)-2,5-diphenyl-2-H-tetrazolium bromide (MTT) was purchased from ApexBio Biotechnology Co., Ltd. (Houston, TX, USA). High quality foetal bovine serum was purchased from Shanghai Excell Biotechnology Inc. (Shanghai, China). Trypsin-EDTA, penicillin-streptomycin and cisplatin were supplied by Solarbio (Shanghai, China). The remaining common reagents were all of analytical grade.

### 3.2. Preparation of the mPEG-PMA Copolymer

Dimethylaminopyridine (DMAP) was used as the catalyst and polyethylene glycol monomethyl ether-polymandelic acid (mPEG-PMA) was synthesized by way of ring-opening polymerization (ROP). First, mandelic acid OCA (Mac-OCA) was synthesized as described in the literature [35,36,37]. Then, mPEG, Mac-OCA and DMAP were added to a dry round-bottom flask at a certain mass ratio. An appropriate amount of dichloromethane (DCM) was used as the solvent and the reaction was stirred at room temperature under nitrogen for 24 h. Most of the DCM was removed under reduced pressure, methyl t-butyl ether and ethanol were added and a white solid was precipitated. The solid was dried under a vacuum at room temperature for 48 hours after solid-liquid separation. The reaction scheme is shown in Figure 10.

### 3.3. Characterization of the mPEG-PMA Copolymer

The proton nuclear magnetic resonance (NMR) spectrum and molecular weight of the samples were determined and analysed using an NMR spectrometer (400 MHz, BRUKER, Karlsruhe, Germany) and gel permeation chromatography (GPC, 7.8 × 300 mm, Waters Styragel, Waters Inc., Milford, MA, USA) to assess the synthesis of the copolymer.

### 3.4. Preparation of PNS-Loaded mPEG-PMA Microspheres

The PNS-loaded mPEG-PMA microspheres were prepared using the emulsion solvent evaporation method. A detailed description of the steps is provided below. Twenty-five milligrams of PNS were dissolved in 1 mL of ethanol. The solution was added to a 10 mL round-bottom flask containing 5 mL of DCM dissolved in 150 mg carrier material in a dropwise manner and stirred at 21,000 rpm in a 0 °C ice bath environment. After the addition, the mixture was continuously stirred at high speed for 3 min to form the first emulsion. The emulsion was continuously added to a small 50 mL beaker containing 20 mL of a 1% aqueous PVA solution in a 0 °C ice bath under high-speed stirring (31,000 rpm) using a syringe in a dropwise manner (completed within 1 min), followed by emulsification through high-speed stirring for 3–5 min to form the double emulsion. The double emulsion was quickly transferred to a 500 mL beaker containing 400 mL of a 5% aqueous isopropanol solution followed by stirring at a low speed (approximately 300 rpm) for 24 h at room temperature to volatilize the organic solvent in the dispersion. This solution was centrifuged at high speed (6,500 rpm) for 10 min, the supernatant was discarded and the white solid at the bottom (microspheres) was collected. The solid was washed 3 times with water and lyophilized at −50 °C for 24 h to obtain a white powder of the microspheres. The blank microspheres were prepared using the same method described above.

### 3.5. Characterization of the Microspheres

The morphology of the mPEG-PMA microspheres loaded with PNS was observed using scanning electron microscopy (SEM, NOVA NANOSEM-450, FEI, Hillsboro, OR, USA) and the particle size distribution of the microspheres was measured using a laser particle size analyser (Mastersizer 3500, Microtrac, Malvern, UK).

The actual encapsulation and drug loading status of all components of PNS in the microspheres were determined using spectrophotometry. The detection wavelength was 287 nm. The absorbance (optical density (OD) value) of the measured series of known concentrations of PNS standard solutions was used as the ordinate and the concentration C (μg/mL) was used as the abscissa to establish the standard curve. The linear range was 2.5–250 μg/mL.

Measurement method: A certain amount of microsphere powder was accurately measured in a centrifuge tube and a small amount of DCM was added to completely dissolve the microsphere-encapsulated material and release the PNS encapsulated in the microsphere. Then, a certain volume of methanol solvent (DCM: MeOH = 1:9) was added to dissolve PNS and precipitate the mPEG-PMA encapsulating material. After centrifugation, the precipitate was discarded and a certain volume of supernatant was removed to determine its OD value. In the experiments, each group of samples was tested three times. The results are presented as mean values and standard deviations. The measured OD value was input into the standard curve to calculate the amount of PNS encapsulated in the microspheres. The formulas used to calculate the entrapment rate (EE%) and the percentage of drug loaded (DL%) in the microspheres were:(1)$$\mathrm{EE}\%\;=\;\frac{{\;M}_{2\;}}{\;M_{0}}\times \;100\%$$
(2)$$\mathrm{DL}\%\;=\;\frac{{\;M}_{2}}{{\;M}_{1}}\times \;100\%$$ where M_0_ is theoretical concentration of PNS loaded (μg), M_1_ is the mass of drug-loaded microspheres (μg) and M_2_ is the measured PNS mass (μg) in the microspheres.

### 3.6. In Vitro Release and Degradation of the Microspheres

Fifty milligrams of microsphere powder were accurately weighed in a 15 mL centrifuge tube and 10 mL of phosphate-buffered saline (PBS buffer solution) was added, followed by shaking in a 37 °C constant temperature shaker. At the set specific time in the experiment, a sample was removed and centrifuged; 1 mL of the release solution was removed from the supernatant and an equal amount of fresh release medium PBS was added. Then, the constant temperature shaking was continued. The PNS concentration in the 1 mL aliquot of the release solution was measured using a microplate reader and the cumulative amount released was calculated using the following formula:(3)$$Q=C_{n}\times V_{t}+V_{s}\sum C_{n-1}$$

Q: cumulative release (µg);Cn: concentration of the release medium at time t (µg /mL)Vt: volume of release medium (mL);Vs: volume of solution obtained from the release medium for testing.

The in vitro release of the microspheres was obtained by plotting the cumulative release percentage Q (%) of the drug-loaded microspheres against the release time t (D).

The conditions for the in vitro degradation of the PNS-loaded mPEG-PMA microspheres were the same as the conditions for the release assay. At set times in the experiment, the sample was centrifuged and the supernatant was used to determine the change in the pH of the degradation medium of the microspheres. The microspheres were washed 3 times with water and lyophilized at −50 °C for 24 h. The lyophilized microspheres were removed and then weighed accurately with an analytical balance to calculate the dry weight loss of the degraded microspheres. Then, 3 mg of the lyophilized microspheres were weighed and dissolved in THF to determine the molecular weight change of the degraded microspheres using GPC. The in vitro degradation curve of the microspheres was obtained by plotting the pH value of the degradation medium, the dry weight of the microspheres and the molecular weight of the microspheres against time, respectively.

### 3.7. Assessment of the Biocompatibility of the Microspheres

The sample was dissolved in distilled water to prepare a suspension with final concentrations of 0.4 μg/mL, 4 μg/mL and 40 μg/mL. In each group, 8 mL of fresh anticoagulant-treated rabbit blood (blood mixed with sodium citrate in a volumetric ratio of 9:1) were diluted with 10 mL of a 0.9% NaCl solution and 200 μL of the sample to be tested were placed in a test tube. Then, 5 mL of 0.9% NaCl were added and incubated in a 37 °C water bath for 30 min. Then, 100 μL of diluted blood were added, followed by gentle mixing and an incubation at 37 °C for 60 min. The positive control comprised 5 mL of deionized water and 100 μL of diluted blood (D = 0.8 ± 0.3) and the negative control comprised 5 mL of the 0.9% NaCl solution and 100 μL of diluted blood. After centrifugation at 800 rpm for 5 min, the supernatant was transferred to a cuvette and the absorbance was measured at a wavelength of 540 nm using a microplate reader. The haemolysis ratio was calculated according to formula (4). When the haemolysis ratio was greater than 5%, the material was considered to exert a haemolysis effect.

(4)$$\mathrm{Hemolytic}\;\mathrm{ratio}\;\left(\%\right)\;=\;\frac{{\mathrm{Abs}}_{\mathrm{sample}}{-\;\mathrm{Abs}}_{\mathrm{negative}\;\mathrm{control}}}{{\mathrm{Abs}}_{\mathrm{positive}\;\mathrm{control}}{-\;\mathrm{Abs}}_{\mathrm{negative}\;\mathrm{control}}}\times \;100\%$$

The microspheres were formulated into suspensions with final concentrations of 0.4 μg/mL, 4 μg/mL and 40 μg/mL. In each group, 200 μL were removed, incubated at 37 °C for 5 min and 50 μL of fresh anticoagulant-treated rabbit blood were injected into the sample. After an incubation at 37 °C for 5 min, 10 μL of an aqueous calcium chloride solution (0.2 mL/L) were injected into the blood, mixed and incubated at constant temperature for 5 min. Then, 12 mL of deionized water were added and the supernatant was removed. The absorbance of the blood:water mixture was recorded at a wavelength of 540 nm using a microplate reader; 12 mL of deionized water containing 50 μL of whole blood served as the control. Each sample was measured 5 times and averaged. The anticoagulant properties of the sample are presented in terms of relative absorbance:(5)$$\mathrm{BCI}=\frac{I_{o}}{\;I_{w}}\;\times 100\%$$ where I_o_ is the relative absorbance of the mixture of blood and calcium chloride after exposure to the sample for a set period of time and I_w_ is the relative absorbance of the blood after mixture with a certain amount of deionized water.

The HL-7702 human normal liver cell line was inoculated into the culture flask and RPMI-1640 medium (containing 10% foetal bovine serum, 100 U/mL penicillin and 100 U/mL streptomycin) was added, followed by culture in a 37 °C incubator with a 5% CO_2_ atmosphere. The cells were grown in a single layer and passaged every 3–5 d using 0.25% trypsin. One hundred eighty microliters of cells in logarithmic growth (5×10^4^ cells/mL) were seeded in 96-well plates and 20 μL of the samples were added and incubated overnight. Three concentration gradients were established with 3 replicate wells per concentration. After 48 h, 20 μL of 5 mg/mL MTT were added to each well and incubated for 4 h. The culture solution was aspirated and 150 μL of DMSO were added to stop the reaction, followed by shaking for 15 min in the shaker. The absorbance at 490 nm was measured using a microplate reader. Cell viability was calculated using the following equation:(6)$$\mathrm{Cell}\;\mathrm{viability}\;\left(\%\right)\;=\frac{{\mathrm{OD}}_{\mathrm{sample}}}{{\mathrm{OD}}_{\mathrm{negative}\;\mathrm{control}}}\times \;100\%$$

### 3.8. Assessment of the Anti-Inflammatory Activity 

The effect of drug-loaded microspheres on xylene-induced ear swelling was measured in 35 mice (male and female) that were randomly selected and randomly divided into 7 groups of 5 animals per group. With the exception of the group treated with aspirin, which was intragastrically administered on the day of the experiments, the other groups were intragastrically administered the compounds once a day for 3 consecutive days. The control group was administered the same volume of a 1% sodium carboxymethylcellulose (CMC-Na) solution. The volume used for intragastric administration in each group was 20 mL/kg bw. Thirty minutes after the last intragastric administration on the day of the experiment, 0.05 mL of xylene was evenly coated on both sides of the right ear of the mice in each group. The left ear was not coated and served as the control. The animals were sacrificed by decapitation 1 h after the induction of inflammation and holes were punched in the same part of the ears using a puncher with diameter of 8 mm. The difference between the weights obtained by punching the same position of the two ears was defined as the degree of swelling.

The effect of drug-loaded microspheres on egg white-induced paw swelling was measured in 35 SD rats (male and female) that were randomly divided into 7 groups of 5 mice per group. With the exception of the aspirin group, which was subjected to only one intragastric treatment on the day of the experiment, the other groups were intragastrically administered the compounds once daily for 3 consecutive days. The control group was administered the same volume of 1% CMC-Na. The volume of the intragastric treatment in each group was 10 mL/kg bw. Thirty minutes after the last intragastric administration on the day of the experiment, 0.1 mL of fresh egg white was subcutaneously injected into the right hind paw of each rat to induce inflammation. The foot volumes before and 1, 2, 3, 4 and 5 h after the induction of inflammation were measured. The difference in the area of the foot before and after the induction of inflammation was used as the swelling rate and the swelling inhibition rate was calculated.

(7)$$\mathrm{Swelling}\;\mathrm{rate}=\left({\mathrm{At}}_{n}-{\mathrm{At}}_{0}\right)/{\mathrm{At}}_{0}$$

(8)$$\mathrm{Inhabition}\;\mathrm{rate}=1-\left(S1_{n}/S2_{n}\right)\times 100\%$$

At_n_: At t = n (*n* = 1, 2, 3, 4, 5) h, the swollen area of the rat toes was observed.At_0_: Initial foot area of rats in each group;S1_n_: At t = n (*n* = 1, 2, 3, 4, 5) h, the swelling rate of toes in the experimental group was higher than that in the control group.S2_n_: At t = n (*n* = 1, 2, 3, 4, 5), the toe swelling rate in the control group was higher than that in the control group.

### 3.9. Assessment of the Anti- Activity 

The experimental procedure for assessing the anti-tumour activity of the drug-loaded microspheres was the same the procedure used for the cytotoxicity experiment. The model was human breast cancer cells. The inhibition rate was calculated using formula (9).

(9)$$\mathrm{Inhibition}\;\mathrm{rate}=1-\frac{{\mathrm{Abs}}_{\mathrm{sample}}}{{\mathrm{Abs}}_{\mathrm{blank}}}\times 100\;\%$$

## 4. Conclusions

A series of copolymers were obtained by ring-opening polymerization of polyethylene glycol monomethyl ether (mPEG) and mandelic acid OCA (Mac-OCA) and a series of mPEG-PMA microspheres containing PNS were prepared by double emulsion method. Through a comprehensive investigation of the appearance, particle size and drug loading and encapsulation efficiencies of the series of drug-loaded microspheres, we have identified the preparation of PNS microspheres that had the best carrier material mPEG_10000_-PMA (1:9). The in vitro release and degradation assays show that the microspheres have good sustained release and degradation properties in vitro and sustainably release the drug for 12 days. The degradation is complete after 60 days of degradation in vitro. The result of biocompatibility experiments displays that the drug-loaded microspheres have good biocompatibility. Anti-inflammatory and anti-tumour experiments show that the drug-loaded microspheres have good anti-inflammatory and anti-tumour activities. Thus, they have broad application prospects.

## Figures and Tables

**Figure 1 molecules-24-02024-f001:**
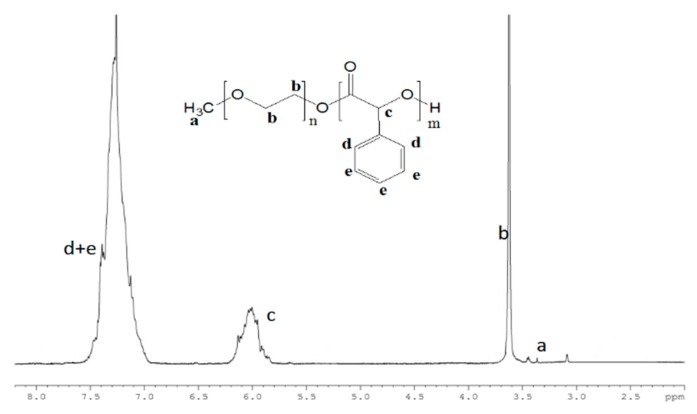
Nuclear magnetic resonance (^1^HNMR) spectra of copolymer Polyethylene glycol monomethyl ether-Polymandelic acid (mPEG-PMA).

**Figure 2 molecules-24-02024-f002:**
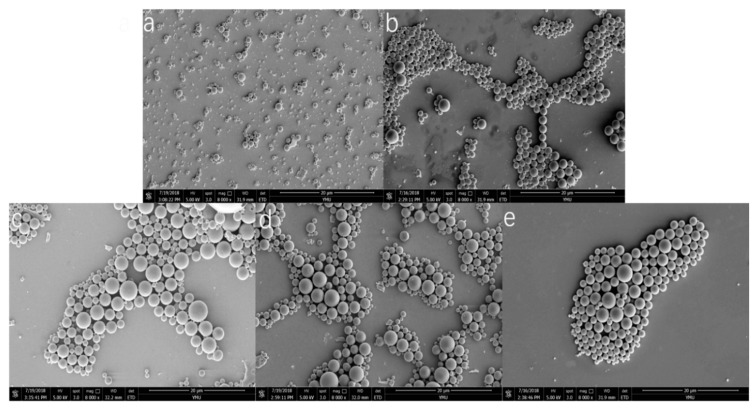
The scanning electron microscopy (SEM) morphology of *Panax notoginseng* saponins (PNS) microspheres. (**a**) mPEG 2k-PMA (1:9), (**b**) mPEG 5k- PMA (1:9), (**c**) mPEG 10k-PMA (1:9), (**d**) mPEG 10k-PMA (1:7), (**e**) mPEG 10k-PMA (1:5). The magnification of SEM image was 8000.

**Figure 3 molecules-24-02024-f003:**
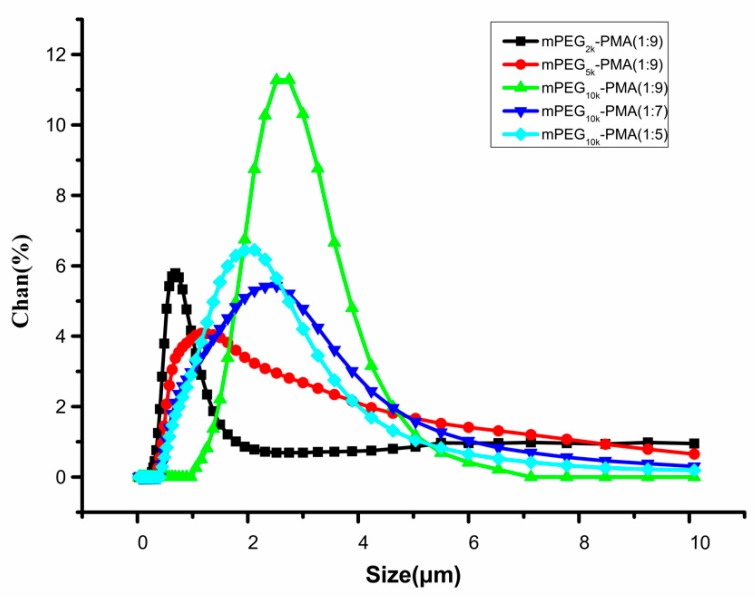
Size distribution of drug-loaded microspheres with different materials.

**Figure 4 molecules-24-02024-f004:**
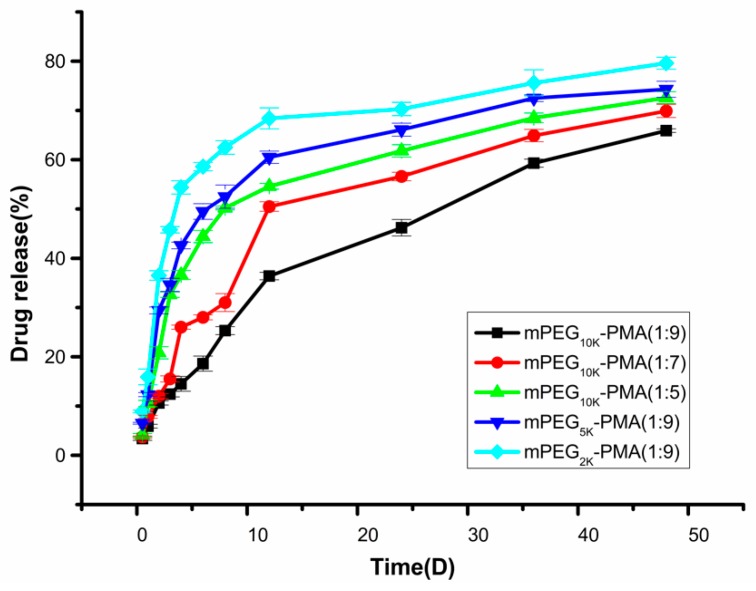
Cumulative release curves of drug-loaded microspheres in vitro.

**Figure 5 molecules-24-02024-f005:**
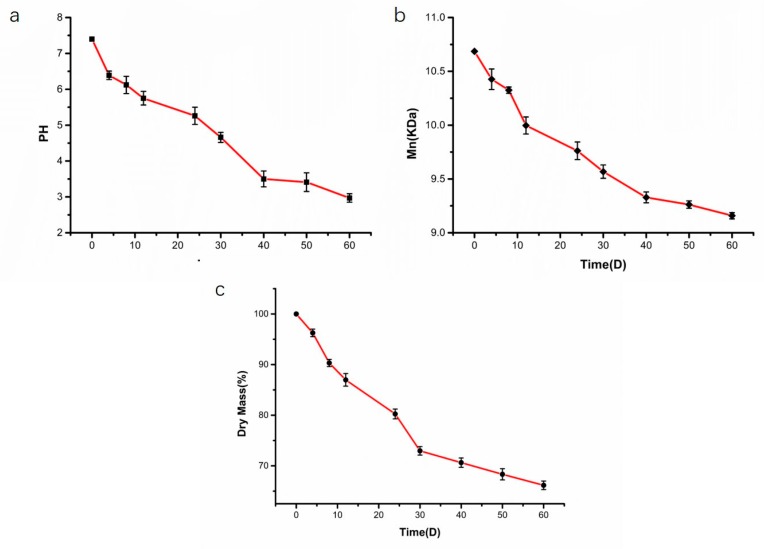
Changes of pH (**a**), Mn (**b**) and dry weight loss (**c**) during degradation of microspheres.

**Figure 6 molecules-24-02024-f006:**
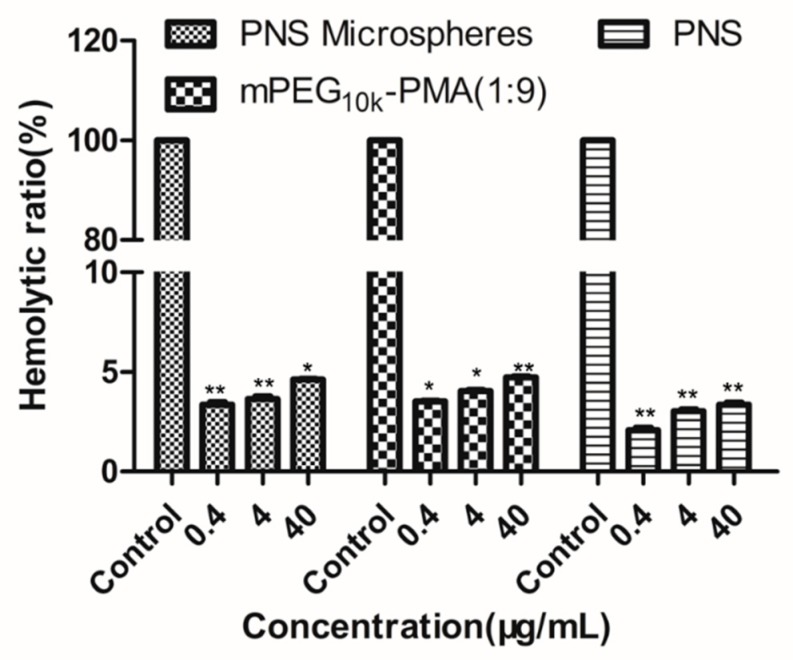
Haemolysis test results of PNS-loaded mPEG_10k_-PMA (1:9) microspheres. * *P* <0.05 and ** *P* < 0.01, compared with control and the whole blood sample with the PNS was used as a control.

**Figure 7 molecules-24-02024-f007:**
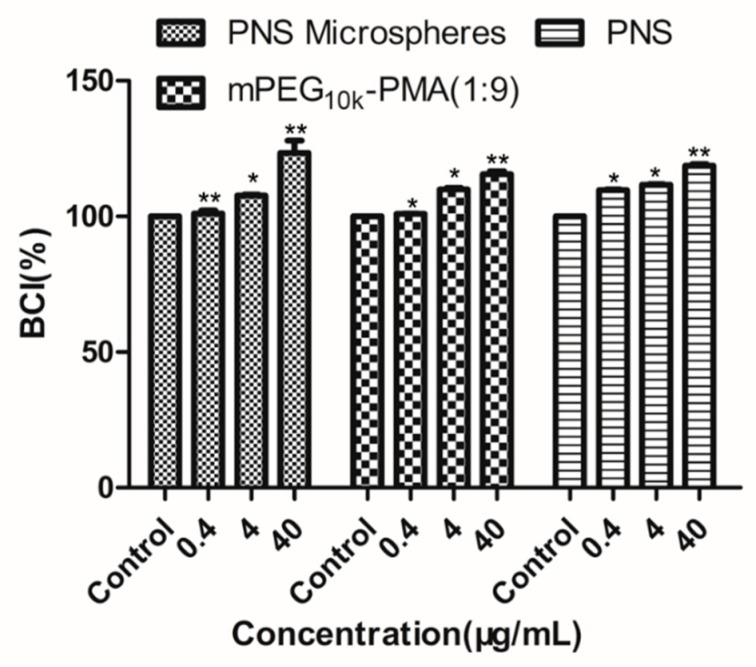
Anticoagulation test results of PNS-loaded microspheres. * *P* < 0.05 and ** *P* < 0.01, compared with control and the whole blood sample with the PNS was used as a control.

**Figure 8 molecules-24-02024-f008:**
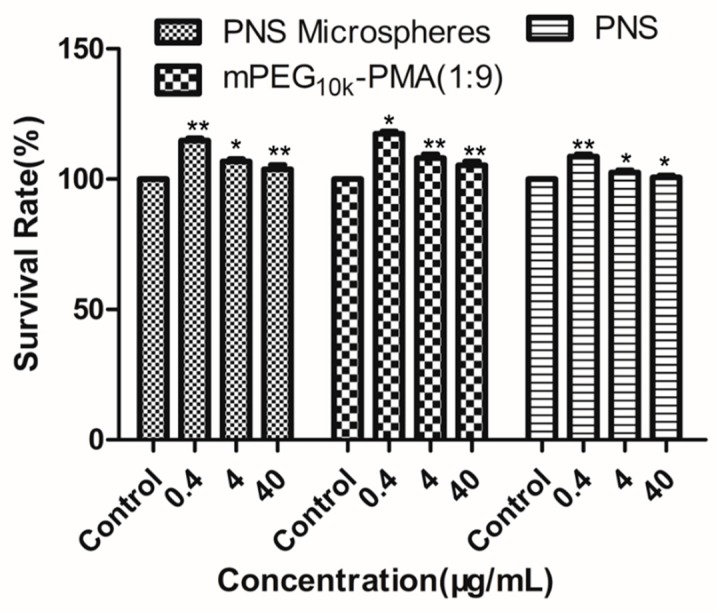
Cytotoxicity test results of PNS-loaded microspheres. * *P* < 0.05 and ** *P* < 0.01, compared with control group. The control group was cells that grew in the same environment without drugs.

**Figure 9 molecules-24-02024-f009:**
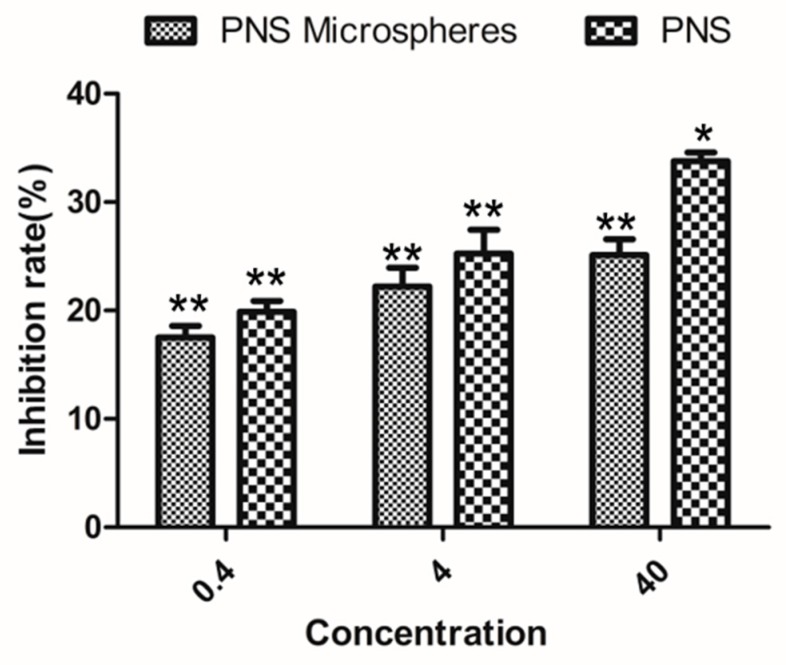
Antitumor activity of PNS and PNS microspheres. * *P <* 0.05 and ** *P <* 0.01 compared with control group. The control group was cells that grew in the same environment without drugs.

**Figure 10 molecules-24-02024-f010:**
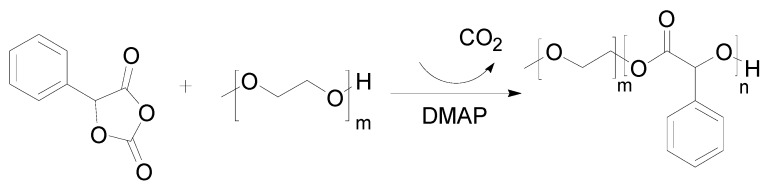
Reaction mechanism of preparing mandelic acid O-carboxyanhydride (Mac-OCA).

**Table 1 molecules-24-02024-t001:** Distribution of the number-average molecular weight (Mn), the weight-average molecular weight (Mw) and polydispersity index of the molecular weight distribution (PDI)of mPEG-PMA Polymer.

Samples	mPEG/Mac-OCA	mPEG	Mn (g/mol)	Mw (g/mol)	PDI
**a**	1:9	2000	2656	2896	1.2041
**b**	1:9	5000	5468	6024	1.1216
**c**	1:9	10000	12596	13523	1.1106
**d**	1:7	10000	11120	11952	1.1947
**e**	1:5	10000	10685	11458	1.1763

Mn: The number-average molecular weight; Mw: The weight-average molecular weight; PDI: polydispersity index of the molecular weight distribution.

**Table 2 molecules-24-02024-t002:** Drug loading and entrapment efficiency of different PNS microspheres.

Samples	mPEG	mPEG/Mac-OCA	DL%	EE%
**a**	2000	1:9	4.12 ± 0.08%	22.69 ± 1.42%
**b**	5000	1:9	6.52 ± 0.12%	35.82 ± 1.57%
**c**	10,000	1:9	8.54 ± 0.16%	47.25 ± 1.64%
**d**	10,000	1:7	7.34 ± 0.13%	40.36 ± 1.82%
**e**	10,000	1:5	6.82 ± 0.14%	37.48 ± 1.25%

EE: encapsulation efficiency. LE: loading efficiency.

**Table 3 molecules-24-02024-t003:** Anti-inflammatory activity of PNS microspheres on xylene-induced auricular swelling in mice.

Groups	Doses (/kg)	Auricular Swelling $\left(\bar{x}\;\pm \;s,\;\mathrm{mg}\right)$	Inhibition Rate (%)
Control group	—	6.17 ± 0.49	—
ASP	10 mg	2.82 ± 0.27	54.29 ± 4.4%
*Panax notoginseng* slices	10 mg	3.52 ± 0.13	45.16 ± 2.65%
PNS	10 mg	3.45 ± 0.27	48.14 ± 4.42%
Microspheres	10 mg	4.08 ± 0.44	33.87 ± 2.06%
	20 mg	3.83 ± 0.18	37.93 ± 2.99%
	40 mg	3.54 ± 0.21	42.63 ± 3.09%

ASP: Aspirin; PNS: *Panax notoginseng* saponins.

**Table 4 molecules-24-02024-t004:** Anti-inflammatory activity of PNS microspheres on toe swelling induced by egg white in rats.

Groups	Doses (/kg)	Inhibitory Rate of Toe Swelling				
		1h	2h	3h	4h	5h
ASP	5 mg	49.16%	65.17%	75.34%	74.58%	72.56%
PNS	5 mg	48.04%	54.48%	70.90%	73.61%	71.50%
*Panax notoginseng* slices	5 mg	22.34%	45.05%	56.01%	58.25%	60.47%
Microspheres	5 mg	32.80%	36.66%	44.77%	49.01%	47.08%
	10 mg	38.23%	41.91%	51.70%	55.09%	54.40%
	20 mg	41.01%	45.55%	53.11%	61.20%	61.06%

ASP: Aspirin; PNS: *Panax notoginseng* saponins.

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
