# Peer review of "Preparation, In Vivo and In Vitro Release of Polyethylene Glycol Monomethyl Ether-Polymandelic Acid Microspheres Loaded Panax Notoginseng Saponins"

_molecules, 2019, doi:10.3390/molecules24102024_

Round 1

Reviewer 1 Report

These workers have prepared and characterized polymeric microspheres loaded with Panax saponins.  They have also demonstrated biocompatibility and bioactivity of the microspheres.  Publication with minor revision is recommended:

  The abstract should include an introductory sentence or two about why this work is important.  The should also include a summary of the results and a concluding sentence.

There are so many abbreviations throughout the manuscript so that it is difficult to follow.  The authors should provide a list of abbreviations.
Additionally, tables should be able to stand alone; please provide definitions of the abbreviations used in the tables as footnotes.

In Figure 2, it is difficult to read the keys on the bottom of each photograph.  Please provide a readable scale on each of the photographs.

Line 184:  "drug-loaded"

The Materials and Methods section should include a sub-section on the humane care and use of laboratory animals along with the approval by the care and use of animals committee.

Author Response

Response to Reviewer 1 Comments

Manuscript ID: molecules-493083

Journal of Molecules

Dear editor,

I have made revisions according to reviewers’ comments. It had been appended below. I hope you can think about my paper again.

Yours sincerely,

Minglong Yuan

Point 1: The abstract should include an introductory sentence or two about why this work is important.  The should also include a summary of the results and a concluding sentence.

Response 1: Thanks for your kind suggestions. Based on your suggestions, I have added relevant sentences to the abstract to introduce the importance of this study and summarize the experimental results. They are the sentence “In order to enrich the types of PNS sustained-release preparations and provide a new research idea for the research and development of traditional Chinese medicine sustained-release formulations.” and the sentence The results showed that the drug-loaded microspheres could be released stably for about 12 days and basically degraded within 60 days. At the same time, the microspheres had good biocompatibility, anti-inflammatory and anti-tumor activities.Please, see line 14-16, 28-30.

Point 2: There are so many abbreviations throughout the manuscript so that it is difficult to follow. The authors should provide a list of abbreviations.
Additionally, tables should be able to stand alone; please provide definitions of the abbreviations used in the tables as footnotes.

Response 2: Thanks for your kind suggestions. The following table is a comparison of abbreviations and full names of professional nouns in the text.

List of abbreviations

Full name

abbreviations

Panax notoginseng saponins

PNS

polyethylene glycol monomethyl ether-polymandelic   acid

mPEG-PMA

Methoxy polyethylene glyco

mPEG

Poly mandelic acid

PMA

nuclear magnetic resonance spectroscopy

1HNMR

gel chromatography

GPC

mandelic   acid O-carboxyanhydride

Mac-OCA

scanning electron microscopy

SEM

Number-average   Molecular Weight

Mn

Mass average molar   mass

Mw

Polymer   dispersity index

PDI

Drug   loading

DL%

Encapsulation   efficiency

EE%

3-(4,5-dimethyl-2-thiazolyl)-2,5-diphenyl-2-H-tetrazolium   bromide

MTT

normal   human liver cells

HL-7702   cells

Dimethylaminopyridine

DMAP

Tetrahydrofuran

THF

dichloromethane

DCM

Sprague-Dawley   rats

SD rats

The   proton nuclear magnetic resonance

NMR

ring-opening   polymerization

ROP

optical density value

OD

phosphate-buffered   saline

PBS

Abs

absorbance

Thanks for your kind suggestions again. Because of your reminder, I understand that those tables should stand alone. But it can only be uploaded for the first time by uploading the relevant pictures and forms of the paper. Now it can not upload the form again. Additionally , in this  journal tables can be left  in the article. Thank you for your understanding.

In addition, according to your suggestion, I have explained the meaning of abbreviations in tables in footnotes below each table in the paper.

Table 1. Distribution of Mn, Mw and PDI of mPEG-PMA Polymer

samples

mPEG/Mac-OCA

mPEG

Mn (g/mol)

Mw (g/mol)

PDI

a

1:9

2000

2656

2896

1.2041

b

1:9

5000

5468

6024

1.1216

c

1:9

10000

12596

13523

1.1106

d

1:7

10000

11120

11952

1.1947

e

1:5

10000

10685

11458

1.1763

Mn: The number-average molecular weight; Mw: The weight-average molecular weight; PDI: polydispersity index of the molecular weight distribution.

Table 2. Drug loading and entrapment efficiency of different PNS microspheres

samples

mPEG

mPEG/Mac-OCA

DL%

EE%

a

2000

1:9

4.12 ± 0.08%

22.69 ± 1.42%

b

5000

1:9

6.52 ± 0.12%

35.82 ± 1.57%

c

10000

1:9

8.54 ± 0.16%

47.25 ± 1.64%

d

10000

1:7

7.34 ± 0.13%

40.36 ± 1.82%

e

10000

1:5

6.82 ± 0.14%

37.48 ± 1.25%

EE: encapsulation efficiency.  LE: loading efficiency

Table 3. Anti-inflammatory activity of PNS microspheres on xylene-induced auricular swelling in mice.

Groups

Doses/kg

Auricular swelling

x(_) ± s, mg

Inhibition rate (%)

Control group

6.17 ± 0.49

ASP

10 mg

2.82 ± 0.27

54.29 ± 4.4%

Panax notoginseng   slices

10 mg

3.52 ± 0.13

45.16 ± 2.65%

PNS

10 mg

3.45 ± 0.27

48.14 ± 4.42%

Microspheres

10 mg

4.08 ± 0.44

33.87 ± 2.06%

20 mg

3.83 ± 0.18

37.93 ± 2.99%

40 mg

3.54 ± 0.21

42.63 ± 3.09%

ASP: Aspirin; PNS: Panax notoginseng saponins

Table 4. Anti-inflammatory activity of PNS microspheres on toe swelling induced by egg white in rats.

Groups

Doses

Inhibitory rate of toe swelling

/kg

1h

2h

3h

4h

5h

ASP

5 mg

49.16%

65.17%

75.34%

74.58%

72.56%

PNS

5 mg

48.04%

54.48%

70.90%

73.61%

71.50%

Panax notoginseng   slices

5 mg

22.34%

45.05%

56.01%

58.25%

60.47%

Microspheres

5 mg

32.80%

36.66%

44.77%

49.01%

47.08%

10 mg

38.23%

41.91%

51.70%

55.09%

54.40%

20 mg

41.01%

45.55%

53.11%

61.20%

61.06%

ASP: Aspirin; PNS: Panax notoginseng saponins

Point 3: In Figure 2, it is difficult to read the keys on the bottom of each photograph.  Please provide a readable scale on each of the photographs.

Response 3: At first, in order to satisfy the typesetting requirements of Figure 2, we reduced the size of the picture, which led to the information below the picture not very clear. Now we have shown the magnification ratio of the picture in the footnote of the picture that " The magnification of SEM image was 8000". Please see line 107-108.

Point 4: "drug-loaded"

Response 4: Thank you very much for pointing out my spelling mistake"drug01loaded". I have corrected it to "drug-loaded".

Point 5: The Materials and Methods section should include a sub-section on the humane care and use of laboratory animals along with the approval by the care and use of animals committee.

Response 5: The following picture shows the license of these animals for animal experiments. According to your suggestion, I added the following sentence in the materials and methods section of the article” All animal experiments were conducted according to the principles of the NIH Guide for the Care and Use of Laboratory Animals and were approved by the ethics committee for laboratory animal care and use of the In-stitute of Materia Medica, CAMS & PUMC.” Please, see line 242-245.

Reviewer 2 Report

Manuscript accept for publication after suggested corrections:

It often happens that the drug during the encapsulation process degrades. The PNS state is very important before and after encapsulation.

1.     Please, show your FTIR spectra of

1.     PNS

2.     mPEG-PMA and

3.     PNS-loaded mPEG-PMA microspheres

Try to comment on the structure of PNS before and after encapsulation.

2.     Show some of the characterization methods of PNS.

3.     Try to explain possible interactions between PNS and mPEG-PMA in PNS-loaded mPEG-PMA microspheres

Author Response

Response to Reviewer 2 Comments

Manuscript ID: molecules-493083

Journal of Molecules

Dear editor,

I have made revisions according to reviewers’ comments. It had been appended below. I hope you can think about my paper again.

Yours sincerely,

Minglong Yuan

Point 1:1.     Please, show your FTIR spectra of

1.     PNS

2.     mPEG-PMA

3.     PNS-loaded mPEG-PMA microspheres

Try to comment on the structure of PNS before and after encapsulation.

Response 1: Thanks for your kind suggestions. Figures A, B and C are the infrared spectra of MPEG-PMA, PNS and microspheres, respectively. It can be seen from the spectra that the characteristic peaks of PNS and mPEG-PMA are at the same position. The signals of PNS and mPEG-PMA may overlap, so that the spectra of mPEG-PMA are similar to those of microspheres.

Spectra A: The broad peak around 3500 cm-1 is attributed to the hydroxyl group in PMA, the peak at 3065 cm-1 is the stretching vibration of CH on the benzene ring in PMA, and the peak at 3035 cm-1 is the stretching vibration of CH in -CH3, the peak at 3000~2850 cm-1 is the stretching vibration of CH in -CH2- in PMA, the strong absorption peak at 1758 cm-1 is the stretching vibration of carbonyl group in the ester group in PMA, and the peak at 1159 cm-1 is the stretching vibration of C-O-C, 733 cm-1 , 696 cm-1 is the characteristic peak of CH bending vibration on the benzene ring in the presence of a monosubstituted group on the benzene ring in PMA.

Spectra B: The broad peak around 3500 cm-1 belongs to the hydroxyl group in PNS, the peak at 2932 cm-1 is the stretching vibration of CH in the alkyl group in PNS, and the peak at 3035 cm-1 is the stretching vibration of CH in -CH2, 1075 cm-1 .The peak at 1039 cm-1 is the stretching vibration of different C-O-C in the PNS.

Spectra C: The peak at 3500 cm-1 is the overlap of the hydroxyl signals in PNS and mPEG-PMA. The peak at 3000-2900 cm-1 is the overlap of the alkyl peak signals in PNS and mPEG-PMA.

The peak at 1200-1100 cm-1 is the overlap of the C-O-C peak signals in PNS and mPEG-PMA.

Point 2: Show some of the characterization methods of PNS.

Response 2: The characterization of total saponins of Panax notoginseng mainly focused on the separation and purification of active components of Panax notoginseng saponins by high performance liquid chromatography (HPLC) and the determination of their contents in Panax notoginseng saponins. The molecular structure of each active monomer has been identified by nuclear magnetic resonance (NMR) in a large number of literatures. This paper mainly focuses on the application of high purity PNS, which does not involve the separation and purification of PNS.

Point 3:  Try to explain possible interactions between PNS and mPEG-PMA in PNS-loaded mPEG-PMA microspheres

Response 3:Panax notoginseng saponins have a certain lipophilicity. MPEG-PMA has a hydrophilic end MPEG and a hydrophobic end PMA. Similar to the similarly compatible principle, the hydrophobic end of the PMA wraps the PNS        to form a "core" inside, and the hydrophilic end of the mPEG forms an outer shell. This results in a typical "core-shell" structure resulting in rounded microspheres. In addition, many of the reactive monomers in PNS contain a large amount of hydroxyl groups, which readily form hydrogen bonds with the carbonyl oxygen atoms in the hydrophobic end PMA, thereby facilitating the hydrophobic end to encapsulate the PNS.

Reviewer 3 Report

Dear Authors,

I appreciate your research focusing on "mPEG10000-PMA (1:9) material is more suitable as a carrier for loading the total saponins of Panax notoginseng" 

would you show the statistical differences in these experiments (hemolytic, anticoagulant, cytotoxicity, anti-inflammatory and anti-tumor experiments) showing how mPEG10000-PMA (1:9) is different from PNS and PNS Microspheres ??? 

Thanks,

LR

Author Response

Response to Reviewer 3 Comments

Manuscript ID: molecules-493083

Journal of Molecules

Dear editor,

I have made revisions according to reviewers’ comments. It had been appended below. I hope you can think about my paper again.

Yours sincerely,

Minglong Yuan

Point 1: I appreciate your research focusing on "mPEG10000-PMA (1:9) material is more suitable as a carrier for loading the total saponins of Panax notoginseng" 

would you show the statistical differences in these experiments (hemolytic, anticoagulant, cytotoxicity, anti-inflammatory and anti-tumor experiments) showing how mPEG10000-PMA (1:9) is different from PNS and PNS Microspheres ??? 

Response 1: Thanks for your kind suggestions. Based on your suggestion, I have made the modifications in the paper as following.

Figure 6. Hemolysis test results of PNS-loaded mPEG10k-PMA (1:9) microspheres. * P0.05 and ** P0.01, compared with control, and the whole blood sample with the PNS was used as a control.

Figure 7. Anticoagulation test results of PNS-loaded microspheres. * P0.05 and ** P0.01, compared with control, and the whole blood sample with the PNS was used as a control.

Figure 8. Cytotoxicity test results of PNS-loaded microspheres. * P0.05 and ** P0.01, compared with control group. The control group was cells that grew in the same environment without drugs.

Figure 9. Antitumor activity of PNS and PNS microspheres. * P0.05 and ** P0.01. compared with control group. The control group was cells that grew in the same environment without drugs.

Response 2: In this paper, MPEG-PMA is found to be a kind of amphiphilic polymer material with good biocompatibility and good degradability. As a traditional Chinese medicine with great application prospects in the field of anti-inflammatory and anti-tumor, PNS has problems such as short half-life and low drug utilization rate in organisms. Using MPEG-PMA as carrier to load PNS can improve the half-life of PNS and improve the bioavailability of PNS, thus avoiding the rapid inactivation of PNS in complex body environment and prolonging the maintenance time of effective concentration. Therefore, it has great advantages in treating inflammation and tumors requiring long-term and high-dose administration.

Round 2

Reviewer 2 Report

It is necessary to explain the results of the IR spectra.

Which groups are assigned peaks at 2932 and 1075 cm-1 (spectra B, PNS). Why are not they on the C spectrum?

Hard evidence is needed that they were obtained PNS-loaded mPEG-PMA microspheres.

Please explain.

Author Response

Response to Reviewer 2 Comments

Manuscript ID: molecules-493083

Journal of Molecules

Dear editor,

I have made revisions according to reviewers’ comments. It had been appended below. I hope you can think about my paper again.

Yours sincerely,

Minglong Yuan

Point 1:It is necessary to explain the results of the IR spectra.

Which groups are assigned peaks at 2932 and 1075 cm-1 (spectra B, PNS). Why are not they on the C spectrum?

Hard evidence is needed that they were obtained PNS-loaded mPEG-PMA microspheres.

Please explain.

Response 1: Thanks for your kind question. It can be seen from the spectra A and B that around the 2922 cm-1 of the spectrum A and the 2932 cm of the spectrum B, they are the stretching vibration peaks of -CH2- in the MPEG-PMA and the PNS, respectively. Additionally, their peak shapes and the positions are the same. Therefore, in the C spectrum, the signals overlap to form a peak at 2915 cm-1 in the C spectrum. The peak positions and the peak shapes of three parts are the same. Similarly, the peak at about 1048 cm-1 in the spectra A is the same as the peak shape at about 1075 in spectrum B, and the peak position is very close to each other. they are both the stretching vibration of the C-O-C ether bond. Therefore, the signals are superimposed to obtain a peak of about 1047 cm-1 in the C spectrum.
